# LEVERAGING SET ASSUMPTION FOR MEMBERSHIP INFERENCE IN LANGUAGE MODELS

## ABSTRACT

Membership Inference (MI) refers to the task of determining whether or not a document is included in the training data of a given model. MI provides an effective post-training alternative for analyzing training datasets when the access to them is restricted, including studying the impact of data choices on downstream performance, detecting copyrighted content in the training sets, and checking for evaluation set contamination. However, black-boxed Language Models (LMs) only providing the loss for the document may not provide a reliable signal for determining memberships. In this work, we leverage the insight that documents sharing certain attributes (e.g., time of creation) are all expected to be in a training set or none of them is, and develop methods that aggregate membership predictions over these documents. We apply our set assumption on five different domains (e.g., Wikipedia, Arxiv), and find that our method enhances prior MI methods by 0.14 in AUROC on average. We further analyze the impact of different language model sizes, training data deduplication, and methods of aggregating membership predictions over sets and find that our method is more effective on undeduplicated and larger models with more documents available in each set and longer sequence sampled for each document, and show our method's robustness against noises in the set assumption under practical settings.

## 1 INTRODUCTION

Language Models (LMs) have demonstrated impressive performance in a range of applications, owing largely to being trained on web-scale texts. However, the increase in the size of the datasets used to train LMs comes at the cost of transparency without releasing access to the datasets, since it becomes increasingly difficult (in the case of open models) or impossible (for closed models) to determine whether a given document is included in the pretraining dataset (Achiam et al., 2023; Team et al., 2023; Touvron et al., 2023).

Membership Inference (MI) refers to the problem of determining whether a document is included in the training data (member) or absent from it (non-member) given only a trained model. Thus, a successful MI approach lets researchers gain knowledge about the presence or absence of training data from black-boxed LMs and better understand the generalization in language models by enabling post-hoc analyses such as: detecting membership of test sets in training data to avoid unfair evaluation of the LMs; inferring the time information of the training data to correlate it with the LMs' downstream performance (K Nylund, 2023); and detecting the presence of copyrighted or licensed content in the training data. Previous MI methods, which we refer as **Individual-MI**, focus on determining the membership of individual documents relying on the logits from a trained LM (Shokri et al., 2017; Yeom et al., 2018; Carlini et al., 2022a). However, the effectiveness of MI is limited by LMs' pretraining data (Shi et al., 2023). We hypothesize that the loss of individual documents often does not provide a strong enough signal for inferring its membership because LMs are often trained on trillions of tokens and only see most of their training documents only once (Touvron et al., 2023).

In this work, we leverage the insight that there exist many sets that are expected to satisfy the **set assumption**: either all documents in the set are present in the training dataset or none of them are, and reform the problem into deciding the membership of entire sets. For example, as shown in Figure 1, Arxiv papers created on the same date are likely to be present or absent from the training data, depending on the data collection date of the LM. We argue that the set assumption are expected

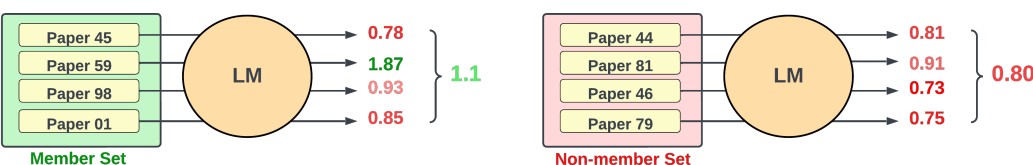

Figure 1: An illustration of the set assumption using Arxiv papers: a set is composed of papers created on the same date, and all paper within a set is likely to be all present or absent from the training data; an LM that collects data on 2020-07-31 will have sets created before that date to be member sets and the ones created after that date to be non-member sets.

Figure 2: An illustration of Set-MI aggregation improves Individual-MI. Individual documents in a member set can have low membership scores (red), making them hard to be distinguished from non-members. By aggregating over documents that share the membership, including documents with strong signals (green), membership scores for members can be more distinguishable from those for non-members.

to hold true in many practical scenarios since documents included in the training datasets are seldom chosen randomly, but based on some inclusion criteria (Albalak et al., 2024), meaning that the sets corresponding to those criteria are entirely present or absent in the training data. For instance, DOLMA (Soldaini et al., 2024), the dataset used to train OLMo (Groeneveld et al., 2024) models, is reported to contain Reddit data up to March 2023, so the set of Reddit posts from April 2023 are all non-members. Based on this assumption, we introduce **Set-MI**, a method that enhances Individual-MI by aggregating signals from sets that satisfy the set assumption. By aggregating over multiple documents, Set-MI can make more accurate predictions and significantly improve the performance of Individual-MI (See Figure 2).

To evaluate Set-MI in practical settings, we construct five benchmarks covering a variety of domains (e.g., Wikipedia, Arxiv) and potential practical applications (e.g., test set contamination, license infringement). We show that Set-MI significantly enhances four Individual-MI methods with an improvement of 0.14 in AUROC on average. To better understand the role that each component of Set-MI plays, we conduct extensive experiments to show the impact of target model sizes, training data deduplication, and the number of tokens that represents each document. Our studies shows that (1) Set-MI improves Individual-MI further when the target model is larger, (2) deduplication in training data makes Set-MI less effective and impact Set-MI more than Individual-MI, and (3) having a longer sequence of tokens to represent the document gives better results. We also conduct a controlled analysis to demonstrate that even in practical scenarios where many sets does not satisfy the set assumption perfectly due to potential training data prepossessing or the existence of duplicated documents, Set-MI is robust against the introduced noise. We do this using evaluation setups with **simulated membership noise**.

Together, Set-MI brings up the limit of MI to inspect the training data of LMs on a practically robust level, provides an approach for any Individual-MI to obtain a significant performance gain by finding documents with shared membership, and provides insights about the contributing factors of the effectiveness of MI on LMs.

## 2 BACKGROUND

Membership inference (MI) in deep learning models is formalized by Shokri et al. (2017) as an adversarial attack to investigate the privacy risk of the models, which is widely used as a component to reconstruct training data (Carlini et al., 2021; Mireshghallah et al., 2022b).[1] While there are work on less difficult settings assuming the access to model parameter (Leino & Fredrikson, 2020) or training loss (Liu et al., 2022), these assumptions limit the application of these methods. We therefore focus on the black-box setting where only the loss score of the target model is available.

MI has been vastly studied with supervised machine learning models (Yeom et al., 2018; Watson et al., 2021; Carlini et al., 2022a). It is also applied to NLP models on pre-training Shi et al. (2023); Duan et al. (2024); Mireshghallah et al. (2022a) and fine-tuning data targeting tasks including text-generation (Mattern et al., 2023; Mahloujifar et al., 2021; Mireshghallah et al., 2022a), translation (Hisamoto et al., 2020), and classification (Shejwalkar et al., 2021). All of these prior work focus on membership on an individual level, as their target of inference is a single sequence instead of a set with a given particular attribute. Jagannatha et al. (2021) proposes using the loss signal as a measure of membership for individual sequences in language models, but they report their results on both an individual and a patient level, where they average the individual sequence signal over all the records belonging to a patient. Our method, however, does not rely on the unique structure of clinical supervised datasets they use and finds natural existing "sets" in the pretraining data originating from the web. We also experiment with more methods rather than simply using the model loss and explore alternative ways to aggregate the signals.

### 2.1 PROBLEM FORMULATION

Given a language model $LM$ trained on pretraining data $\mathcal{T}$ (where $\mathcal{T}$ is typically unknown), and a dataset $\mathcal{D} = \{d_1, d_2, \cdots, d_n\}$ as a collection of documents, the goal of membership inference is to predict whether each document $d_i \in \mathcal{D}$ is a member of $\mathcal{T}$ or not, given only the model's probability over a document $LM(d_i)$ and no access to $\mathcal{T}$. MI methods typically define a function $\mathcal{F}$ that outputs a score $\mathcal{F}(LM, d_i)$, representing the membership of $d_i$ in $\mathcal{T}$. In Section 2.2, we provide an overview of various instances of scoring functions used in prior work.

### 2.2 INDIVIDUAL-MI

We discuss MI methods in prior research, which we refer to as **Individual-MI**, that can be applied to infer the membership of an **individual** document in the pretraining data. In Section 3, we show how our method can build upon each Individual-MI method. For all methods, we randomly select a sequence of tokens $t_1, t_2, \cdots, t_m$ to represent a document $d_i$, and we denote the probability of a token $t_i$ assigned by $LM$ as $LM(t)$.

**Loss Attack** uses $\frac{1}{m} \sum_{i=1}^{m} LM(t_i) = \mathcal{F}(LM, d_i)$, as $LM$ is likely to assign a high probability score to a document that it has seen during training. However, this attack does not consider the complexity of $d_i$, e.g., natural conversations have intrinsically higher probability scores than poetry regardless of the membership (Shokri et al., 2017; Carlini et al., 2022a).

**Likelihood Ratio Attack (LiRA; Carlini et al. (2022a))** proposes to use an additional LM $LM_r$, called a reference model, and defines $\mathcal{F}(LM, d) = \frac{1}{m} \sum_{i=1}^{m} LM(t_i) / \frac{1}{m} \sum_{i=1}^{m} LM_r(t_i)$. The normalization term considers the intrinsic complexity of a document $d$. Ideally, a reference model should be trained on the data that is from the same distribution as $LM$'s training data but with no overlap, so that an unseen document can be distinguished from a complex document. In practice, finding such a reference model is difficult as there is a high overlap between different LMs' training datasets (e.g., English Wikipedia).

**MIN-K% PROB (Shi et al., 2023)** uses the K% of the tokens with the lowest probabilities, denoted as $Lk$. It then defines $\mathcal{F}(LM, d) = \sum_{t \in Lk} LM(t) / |Lk|$. It is designed to remove the effect of tokens that are high-frequency in general web text and thus will have high probabilities regardless of their membership.

---

[1] It was originally referred as "membership inference attack," but we drop the term, "attack," because the intended use for our method is for research purpose and not adversarial.

**zlib entropy (Carlini et al., 2021)** computes a ratio between the size of a document in bits after and before the zlib compression (Gailly & Adler), denoted as $\mathcal{E}(d)$. It then defines $\mathcal{F}(LM, d) = \frac{1}{m} \sum_{i=1}^{m} LM(t_i)/\mathcal{E}(d)$. It shares the motivation with LiRA and uses $\mathcal{E}(d)$ as a measure of the intrinsic simplicity of a document $d$.

## 3 METHOD

Predicting membership of individual documents in large LM pretraining datasets using Individual-MI may not often be effective since the signal obtained from each document may not be strong enough. We introduce **Set-MI**, a method to augment existing Individual-MI methods by using the set membership assumption of individual documents and aggregating their predictions. We formally introduce our set assumption below, followed by the modification to existing MI methods using our Set-MI aggregation.

**Set assumption.** Following the notation from Section 2.1, for MI of a document collection $\mathcal{D}$ against a training set $\mathcal{T}$, we assume that the collection of documents $\mathcal{D}$ can be split, based on their metadata, into multiple disjoint sets of documents $S_i$

$$\mathcal{D} = \bigcup_{i=1}^{k} S_i \text{ and } S_i \cap S_j = \emptyset \text{ for } S_i \, , S_j \in D, i \neq j$$

and that either all the elements in any set $s_i$ are present in $\mathcal{T}$ or none of them is in $\mathcal{T}$, formally

$$\mathcal{M}(d_x, \mathcal{T}) = \mathcal{M}(d_y, \mathcal{T}) \forall d_x, d_y \in s_i$$

where $\mathcal{M}(d, \mathcal{T}) = 1$ iff $d \in \mathcal{T}$ and 0 otherwise.

**Set-MI aggregation.** We define the aggregated membership score over a set as

$$\mathcal{F}(LM, s) = \frac{1}{|s'|} \sum_{\substack{d \in s' \\ s' \subseteq s}} \mathcal{F}(LM, d).$$

where we simply take the average of all the documents in the set as the final score. We optionally select a subset of documents, $s'$ to aggregate the scores instead of the whole set. Finally, we assign the aggregated score from the set to every element within the set, so that the score is directly comparable with previous methods:

$$\forall_{d \in s} \mathcal{F}(LM, d) = \mathcal{F}(LM, s).$$

**Practicality of the Set Assumption.** We can generally expect many naturally-occurring sets that satisfy the set assumption at different levels of granularity in training datasets, as the training data are often chosen based on some inclusion criteria (Albalak et al., 2024), which can be shared across different documents. For example, articles published on the same date, documents with the same license, and instances from an existing smaller dataset all compose sets that are likely to be entirely present in the trainings sets of LMs. Figure 1 shows an example where sets composed by Arxiv papers created on the same date satisfies the set assumption. However, practical factors like LMs' filtering procedure on its training data and the potential duplication across different documents (Albalak et al., 2024; Elazar et al., 2023) might cause the set assumption to not hold in some cases. We argue that even in such cases, Set-MI can still yield significant benefits. Firstly, we provide evidence that our approach is robust to membership noise (see Section 6 where we evaluate under simulated membership noise). Secondly, the applications of Set-MI are not affected by the deviation between the perfect set assumption and the practice, including revealing the time cutoff of LMs' training data collection, checking for potential dataset contamination post-hoc, learning the composition of the alignment data, etc. Based on these insights, we construct benchmarks in the following Section 4.

## 4 EVALUATION BENCHMARK CONSTRUCTION

We construct evaluation benchmarks covering a range of domains and use cases for membership inference. Each benchmark includes a collection of documents $\mathcal{D}$, and sets $s_1...s_k$ that are likely to

| Domain | Statistics | | Examples for set assumption |
|---|---|---|---|
| | # Sets | # Docs | |
| Wikipedia | 1,000 | 100,000 | Articles posted on 2020-01-01, articles posted on 2020-01-02, ... |
| Arxiv | 1,000 | 100,000 | Papers posted on 2020-01-01, papers posted on 2020-01-02, ... |
| Languages | 200 | 20,000 | English, French, Spanish, ... |
| License | 190 | 19,000 | All datasets under Public Domain, all datasets under CC-BY, ... |
| Instructions | 130 | 13,000 | ShareGPT, GPT4-Alpaca, Code-Alpaca ... |

Table 1: List of the benchmarks statistics used in this work, along with example sets.

satisfy the set assumption described in Section 3. The ground-truth membership of each document in $\mathcal{D}$ depends on the target LM, which we will discuss in Section 5. To our knowledge, the MI benchmarks we construct are the first ones that are based on sets of documents, and are the most diverse among LM's MI benchmark with five different domains.

We discuss the motivation and data collection procedure for each of them below (see their statistics in Table 1).

**Wikipedia.** In Wikipedia, articles created on the same date are likely to be all included or excluded from the training data of an LM. Therefore, we can define each $s_i$ with a set of articles created on the same date. To construct the dataset, we collect articles in 2023-11-20 Wikipedia English dump from Wikimedia grouped by their creation date from 2001-01-21 to 2023-10-27. We subsample 100 sets with 100 documents per set.

**Arxiv.** Similarly, we can define each $s_i$ with a set of Arxiv papers posted on the same date. To construct the dataset, we collect Arxiv documents from Redpajama (Computer, 2023) grouped by publication dates from 1991-08-21 to 2023-03-07. We subsample 100 sets with 100 documents per set.

**Language.** Wikipedia contains articles in different languages, and models often choose certain languages to include in their training data and ignore the rest. We therefore collect Wikipedia articles in 20 different languages from Redpajama and define articles of each language as a set. For robust evaluation, we subsample 1,000 documents for each set and divide them into 10 smaller sets, resulting in 130 sets with 100 documents per set.

**License.** As concerns surrounding the use of copyrighted data in training LMs continue to raise (Henderson et al., 2023), recent work has selected training data based on their license information. Different models may choose different criteria for determining which license can be included in their training sets, from a highly restrictive approach where only public domain data can be used, to a more permissive option that includes permissively-licensed code or creative commons datasets. Datasets with the same license are likely to be either all included or all excluded from the training data. To study such a setting, we collect a combination of 15 datasets from the Open License Corpus (Min et al., 2023), along with an additional 4 datasets from the Pile, categorized according to their licenses. For robust evaluation, we subsample 1,000 documents for each set and divide them into 10 smaller sets, resulting in 130 sets with 100 documents per set.

**Instructions.** There has been much work collecting large-scale NLP datasets with instructions and training the LM to follow the instructions to perform tasks. In these models, each dataset, consisting of a set of input and output pairs, essentially shares membership. To study such a setting, we collect a collection of 13 instructing tuning datasets sourced from both Tulu-v1 mix (Wang et al., 2023) and Tulu-v2 mix (Ivison et al., 2023), grouped by its associated dataset. For robust evaluation, we subsample 1,000 documents for each set and divide them into 10 smaller sets, resulting in 130 sets with 100 documents per set.

## 5 EXPERIMENTS

We present an evaluation of the effect of aggregating membership predictions using Set-MI. We compare Individual-MI and their corresponding Set-MI enhanced variants in multiple settings, where

| Method | Loss Attack | | LiRA | | Min-K% PROB | | zlib entropy | |
|---|---|---|---|---|---|---|---|---|
| | Ind-MI | Set-MI | Ind-MI | Set-MI | Ind-MI | Set-MI | Ind-MI | Set-MI |
| Wikipedia | 0.524 | **0.575** | 0.581 | **0.859** | 0.545 | **0.749** | 0.519 | **0.566** |
| Arxiv | 0.576 | **0.938** | 0.508 | **0.576** | 0.590 | **0.954** | 0.560 | **0.827** |
| Languages | 0.836 | **0.960** | 0.908 | **1.000** | 0.740 | **0.760** | 0.673 | **0.733** |
| License | 0.706 | **0.758** | 0.761 | **0.859** | 0.810 | **0.913** | 0.647 | **0.674** |
| Instructions | 0.596 | **0.786** | 0.612 | **0.857** | 1.000 | **1.000** | **0.458** | 0.429 |
| Average | 0.620 | **0.799** | 0.637 | **0.835** | 0.632 | **0.851** | 0.556 | **0.638** |

Table 2: Main results of Individual-MI and Set-MI on five benchmarks we construct. The results for Wikipedia, Arxiv, and License are the average of multiple models (see Appendix C, Appendix D, and Appendix E for the full results). Ind-MI refers to Individual-MI. **Set-MI significantly outperforms Individual-MI on most settings.**

we vary key aspects including the language models, the training data, and the document collections whose membership is being inferred.

All experiments in this section aggregate membership scores over the entire set. That is, $s' = s$, following the notation used in Section 3. We randomly subsample 1,024 tokens as the representation for each document for our main experiment. See the choices for reference models we use for LiRA and more implementation details in Appendix A.

**Target LMs.** We choose a collection of primary target LMs for each dataset and define the ground truth membership based on the known metadata about the target LMs. Statistics of members and non-members for each dataset are provided in Appendix B.

For **Wikipedia** and **Arxiv**, we use the Pythia models (Biderman et al., 2023) and GPT-Neo models (Black et al., 2021) as our primary target LMs, as these models' training data, the Pile (Gao et al., 2020), is publicly available and thus we can use it to determine the ground-truth membership of the data. We label the ground-truth membership of each document based on whether their creation date is before the data collection day of the Pile, i.e., 2020-03-01 for Wikipedia, and 2020-08-01 for Arxiv. We primarily report results with Pythia-12B-dedup, and report ablations with varying sizes of LMs and compare LMs with and without deduplication.

For **Language**, we use Bloom-7B (Scao et al., 2022) as our primary target LM, and use the Wikipedia languages that were used to train the model on to determine the ground truth membership of the data.

For **License**, we use all three variants of SILO models, SILO-PD-1.3B, SILO-PDSW-1.3B, SILO-PDSWBY-1.3B (Min et al., 2023), as target LMs, and use the license category that the authors used for training their LM to determine the ground truth membership of the data.

For **Instructions**, we use Tulu-v1 (Wang et al., 2023) as a primary target LM, and use the dataset the authors used for training to determine the ground truth membership of the data.

### 5.1 MAIN RESULTS: EFFECT OF SET-MI

Results of Set-MI compared to Individual-MI on the benchmarks from Section 4 are shown in Table 2. We average the results of two target models for the Wikipedia and Arxiv domains (see full results in Appendix C and Appendix D) and average the results of three target models for the License domain (see full results in Appendix E). We use a single target model for Language and Instructions.

First, we find that the AUROC scores from Individual-MI are often less than 0.1 higher than the random baseline of 0.5, indicating the difficulty of membership inference and the lack of a strong signal from these methods. Set-MI significantly improves over its Individual-MI counterparts with a gain in AUROC score of 0.14 on average, indicating that our set aggregation is effective in improving any Individual-MI method.

Intuitively, the performance of Set-MI depends on the quality of the based-on Individual-MI method, and poor Individual-MI could lead to even worse Set-MI. Specifically, we find a correlation of 0.824 with a $p$-value of 0.0002 between the performance of Individual-MI and the performance of

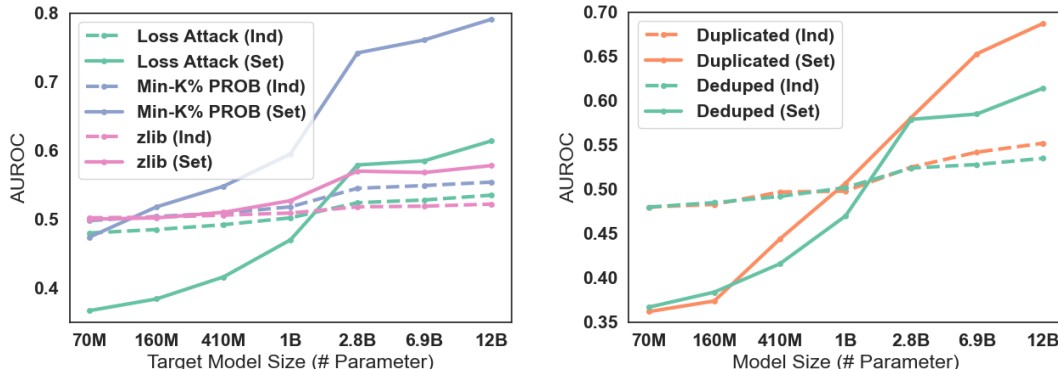

Figure 3: **Left: Effect of Target Model Size.** Performance of Individual-MI ("Ind") and Set-MI ("Set") with target models of different sizes. Performance increases as the target model sizes increase, consistently across all MI methods. Increase in MI scores with larger target models is more significant with Set-MI than with Individual-MI. **Right: Effect of Deduplication.** Performance with target models trained on the original Pile ("Duplicated") and the deduplicated Pile ("Deduped"). The gaps between Duplicated and Dedupped are larger with Set-MI than with Individual-MI.

Set-MIThis suggests that future improvements to Individual-MI has the potential to further improve Set-MI.

## 5.2 Effect of Target Model Size

How does the size of the model affect the effectiveness of Set-MI? Prior work (Carlini et al., 2021; 2022b) have shown that larger models tend to memorize more; does that directly benefit Set-MI? To answer these questions, we study Set-MI aggregation over three Individual-MI methods on Wikipedia with Pythia models of sizes varying from 70M to 12B.

Figure 3 (left) demonstrates that scores of both Individual-MI and Set-MI increase as the target model size increases, confirming the findings from Carlini et al. (2021; 2022b). This trend is significantly more evident with Set-MI than with Individual-MI. This suggests that as the model gets larger, its memorization of the training data becomes stronger, which Set-MI can better exploit.

It is also worth noting that when target model is small, Set-MI may perform worse than Individual-MI. This is because Individual-MI is not effective (AUROC under 0.5) with these small models, making Set-MI aggregating opposite signals as discussed in Section 5.1.

## 5.3 Effect of Deduplication

Lee et al. (2021) and Kandpal et al. (2022) have shown that that MI is less effective on LMs with deduplicated data than the duplicated ones. To confirm their findings with Set-MI, we compare the performance of Loss Attack on Wikipedia between Pythia models trained on the original Pile ("Duplicated") and models trained on the deduplicated Pile ("Deduped"), with varying sizes from 70M to 12B.

We report the results in Figure 3 (right). We do not see clear differences in MI performance between the Duplicated models and Deduped models with Individual-MI. Set-MI's performance is substantially higher with the Duplicated models than with the Deduped models with model sizes greater than 410M, despite not showing a difference with 70M and 160M models, This suggests Set-MI also becomes less effective in exploiting models' memorization against the deduplication on models' the training data.

## 5.4 Effect of Document Length

As introduced in Section 2.2, each document is represented as a random segment of length 1024. We vary the length of this segment from 16 to 2,048, and see how that affects Set-MI.

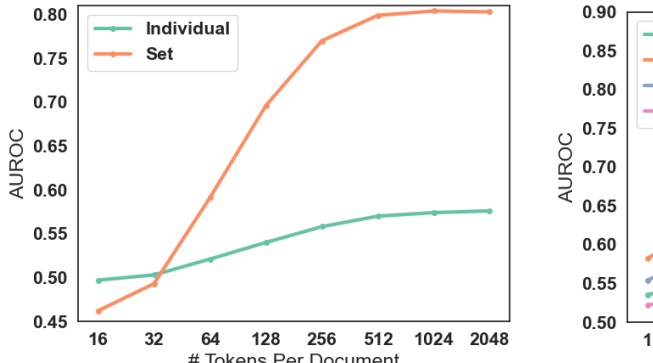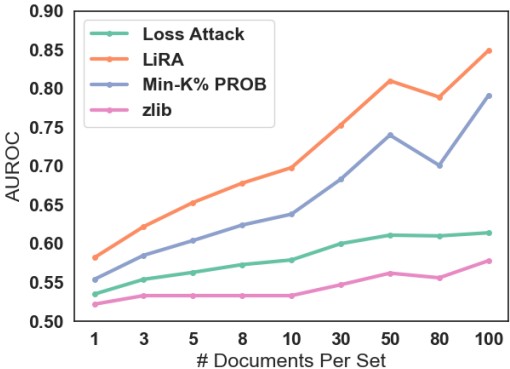

Figure 4: **Left: Effect of Document Length.** Performance of LiRA. The $x$-axis indicates # tokens we sample from each documents, varying from 16 to 2048. The performance gain from Set-MI increases as the # tokens increases. **Right: Effect of Available Documents Per Set.** The $x$-axis indicates # available documents within each set, varying from 1 to 100. Note that Set-MI is equivalent to Individual-MI when there is only one document available. Set-MI gains better performance as # documents within each set increases.

Figure 4 (left) reports results of LiRA on Wikipedia. MI performance increases with longer samples, suggesting that using longer samples for each document provides a stronger signal for MI. The gap between Set-MI and Individual-MI increases as the length of the sample from each document increases, likely because Set-MI exploits signals from longer samples better. Note that the performance becomes saturated as the performance difference is small between performance with 256 tokens v.s. 2,048 tokens.

## 5.5 EFFECT OF AVAILABLE DOCUMENTS PER SET

In practical applications of Set-MI, the number of available documents in each set may be limited. To quantify the impact of the size of the set, we evaluate Set-MI on Wikipedia with varying sizes of sets, from 1 to 100. Note that when the set size is 1, Set-MI is equivalent to Individual-MI.

Results are shown in Figure 4 (right). The performance of Set-MI increases as the number of available documents increases. This is likely because aggregation over the bigger set provides stronger signals about the membership of the sets. We also find the set size does not have to be very big in order to benefit from Set-MI, e.g., the set size of 3 already provides significant gains over Individual-MI (the set size of 1) .

## 6 ROBUSTNESS OF SET AGGREGATION

As discussed in Section 3, due to practical issues like filtering and duplication, the set assumption can rarely hold perfectly in real-world. We thus conduct an carefully controlled analysis to explicitly explore how the noise in Set-MI datasets, which breaks the set assumption, impacts the performance of aggregation methods, and we provide insights about which aggregation method to choose based on prior knowledge about potential types of noise.

We start off by constructing a clean version of Wikipedia, targeting at the deduplicated version of the Pythia 2.8B model, where all documents' memberships are correctly labeled according to their 13-gram overlap with the model's training data (i.e., the Pile). This simulates a ideal dataset where our set assumption perfectly holds. To simulate noise, we replace a portion of documents in a set with documents of the opposite membership. We investigate the performance of Loss Attack under the settings where the noise is included in either member or non-member sets with noise ratios ranging from 0.0 to 0.9, as well as noise in both sets with ratios ranging from 0.0 to 0.5. We compare three aggregation methods: 1) **MAX**: averaging the highest $30\%$ of membership scores in each set, 2) **MIN**: averaging the lowest $30\%$ of membership scores in each set, and 3) **FULL**: average all the membership scores in each set.

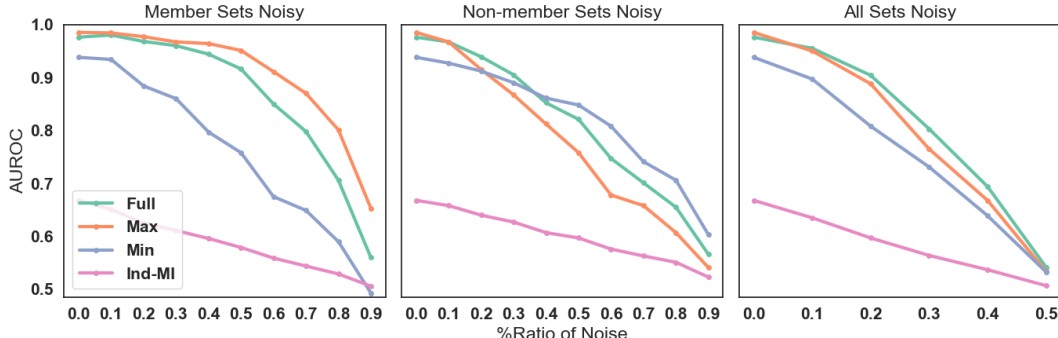

Figure 5: **Aggregation Over Noisy Set.** Performance of three aggregation methods under the settings where there are increasing ratio of noisy documents existing in (1) **left**: member sets, (2) **middle**: non-member sets, and (3) **right**: both member and non-member sets. All three methods outperform Individual-MI under noisy settings.

Figure 5 shows the performance of Set-MI with the three aggregation methods in the three different scenarios. When only the member sets have noise, MAX is the most resistant aggregation method. This is because MAX focuses on the documents with higher scores in member sets, which are less likely to be affected by the noise. MIN, on the other hand, focuses on the documents in the member set with the lowest scores, which are heavily affected by the noise. When noise is introduced into the non-member sets, similarly yet oppositely, MIN is the most resistant and MAX is the least resistant. When both member and non-member sets contain noise, FULL tends to perform the best because MAX and MIN are influenced by noise either in member sets or non-members sets. All three aggregations significantly outperform Individual-MI in all three settings. In practice, we recommend users select the best aggregation based on their prior knowledge about the noise in the set of their interest.

## 7 CONCLUSION

Membership inference (MI) lets researchers infer the composition of training datasets from trained language models and trace specific downstream behaviors back to the training data of these models. However, the performance of existing loss-based MI methods is not good enough for practical use, likely because the loss from a trained model does not provide sufficiently strong signals to distinguish members from non-members. This paper introduces Set-MI, a new MI method that identifies a set of documents that shares the membership (e.g., arXiv papers posted on the same date) and aggregates MI scores over documents within the set. Set-MI is orthogonal to previous MI methods (Individual-MI) and can added on top of them to improve their performance. On five different new benchmarks, Set-MI leads to significant improvements in AUROC scores, consistently across four previous MI methods. Our analyses demonstrate that Set-MI effectively exploits stronger membership signals from the target models, e.g., when target models memorize more due to their larger size and the duplication in the training data. Our study further shows that Set-MI is still robust against the noise potentially introduced by sets with imperfect set assumptions.

Our work makes an assumption that the metadata about the dataset of interest ($\mathcal{D}$) is available, enabling the identification of sets that satisfy the set assumption. We highlight that this may not always be the case in practical MI scenarios, and leave relaxing this assumption for future work.

### ETHICAL CONSIDERATIONS

Although our work deliberately drops the "attack" term from "Membership Inference Attack" as our motivation is to provide an alternative method to learn about LM pretraining data composition for research or copyright purposes, it can be used in an adversarial way to threaten users' privacy as prior work have shown. However, due to the nature of our method, we argue that it is difficult to apply our method to enhance adversarial attacks without access or metadata to user's private information.

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

## A  Implementation Details

**Membership Definition.**  Because of the preprocessing that LMs usually perform on their data, the member data we label in our datasets may not be verbatim to the data the target models are trained on. We consider it as an inherent challenge for MIA on LMs, and choose to keep this realistic setting.

**Set Size.**  We subsampled 100 documents from each set as available documents. See Section 5.5 for a study on the effect of the number of documents in each set.

**Context Length.**  We use 512 as the context length that represents each document. We later discuss the effect of the context length in Section 5.4.

## B  Benchmark Statistics

We list statistics and reference models for LiRA of the benchmarks we construct in Table 3.

| Domain | Member | | Non-member | | Reference Model for LiRA |
|---|---|---|---|---|---|
| | # Sets | # Docs | # Sets | # Docs | |
| Wikipedia | 770 | 77,000 | 230 | 23,000 | Pythia-160 |
| Arxiv | 926 | 92,600 | 74 | 7,400 | Pythia-160 |
| Languages | 90 | 5,000 | 150 | 15,000 | OPT-125M |
| License-PD | 40 | 4,000 | 150 | 15,000 | OPT-125M |
| License-PDSW | 90 | 5,000 | 150 | 15,000 | OPT-125M |
| License-PDSWBY | 150 | 15,000 | 40 | 4,000 | OPT-125M |
| Instructions | 70 | 7,000 | 60 | 6,000 | LLama2-13B |

Table 3: List of the statistics and reference model for LiRA of benchmarks we constructed. Note that numbers for members and non-members vary for License based on the target models.

## C  Full Results in the Wikipedia Domain

Apart from the average results in the Wikipedia domain shown in Section 5.1, we show full results for both target models, Pythia-12B-dedup and GPT-Neo-2.7B, in Table 4. Consistent with findings from Table 2, Set-MI consistently outperforms Individual-MI.

| Target Model | Loss Attack | | LiRA | | Min-K% PROB | | zlib entropy | |
|---|---|---|---|---|---|---|---|---|
| | Ind-MI | Set-MI | Ind-MI | Set-MI | Ind-MI | Set-MI | Ind-MI | Set-MI |
| Pythia-12B-dedup | 0.535 | 0.614 | 0.582 | 0.849 | 0.554 | 0.791 | 0.522 | 0.578 |
| GPT-Neo-2.7B | 0.514 | 0.536 | 0.580 | 0.870 | 0.535 | 0.707 | 0.516 | 0.555 |
| Average | 0.524 | **0.575** | 0.581 | **0.859** | 0.545 | **0.749** | 0.519 | **0.566** |

Table 4: Results of Individual-MI and Set-MI on the Wikipedia benchmark with all three target models. Ind-MI refers to Individual-MI.

## D  Full Results in the Arxiv Domain

Apart from the average results in the Arxiv domain shown in Section 5.1, we show full results for both target models, Pythia-12B-dedup and GPT-Neo-2.7B, in Table 5. Consistent with findings from Table 2, Set-MI consistently outperforms Individual-MI.

| Target Model | Loss Attack | | LiRA | | Min-K% PROB | | zlib entropy | |
|---|---|---|---|---|---|---|---|---|
| | Ind-MI | Set-MI | Ind-MI | Set-MI | Ind-MI | Set-MI | Ind-MI | Set-MI |
| Pythia-12B-dedup | 0.579 | 0.946 | 0.516 | 0.634 | 0.594 | 0.966 | 0.562 | 0.831 |
| GPT-Neo-2.7B | 0.574 | 0.930 | 0.501 | 0.518 | 0.586 | 0.942 | 0.559 | 0.822 |
| Average | 0.576 | **0.938** | 0.508 | **0.576** | 0.590 | **0.954** | 0.560 | **0.827** |

Table 5: Results of Individual-MI and Set-MI on the Arxiv benchmark with all three target models. Ind-MI refers to Individual-MI.

## E    FULL RESULTS IN THE LICENSE DOMAIN

Apart from the average results in the License domain shown in Section 5.1, we show full results for all three target models, SILO-PD-1.3B, SILO-PDSW-1.3B, and SILO-PDSWBY-1.3B, in Table 6. Consistent with findings from Table 2, Set-MI consistently outperforms Individual-MI.

| Target Model | Loss Attack | | LiRA | | Min-K% PROB | | zlib entropy | |
|---|---|---|---|---|---|---|---|---|
| | Ind-MI | Set-MI | Ind-MI | Set-MI | Ind-MI | Set-MI | Ind-MI | Set-MI |
| SILO-PD-1.3B | 0.710 | 0.732 | 0.775 | 0.821 | 0.860 | 0.964 | 0.637 | 0.571 |
| SILO-PDSW-1.3B | 0.772 | 0.852 | 0.838 | 1.000 | 0.745 | 0.813 | 0.682 | 0.716 |
| SILO-PDSWBY-1.3B | 0.637 | 0.689 | 0.669 | 0.756 | 0.824 | 0.962 | 0.623 | 0.733 |
| Average | 0.706 | **0.758** | 0.761 | **0.859** | 0.810 | **0.913** | 0.647 | **0.674** |

Table 6: Results of Individual-MI and Set-MI on the License benchmark with all three target models. Ind-MI refers to Individual-MI.

## F    EFFECT OF THE REFERENCE MODEL FOR LiRA

LiRA is a particularly effective method for MI, and the choice of reference model is extremely important. In this ablation, we aim to characterize a good reference model for LiRA in language modeling when given a target model. To study this, targetting at Pythia-12-deduped trained on the Pile, we evaluate the performance of Set-MI with LiRA on Wikipedia using reference models with different sizes from Pythia family and GPT-Neo family (Black et al., 2021) trained on the Pile, OPT family (Zhang et al., 2022) trained on RoBERTa+the Pile+PushShift.io Reddit, and GPT-2 family (Radford et al., 2019) trained on WebCrawl, a filtered corpus from the CommonCrawl. Results are shown in Figure 6.

**Good reference models represent the training data distribution of the target model.**    LiRA normalizes the calculated membership scores using a reference model so that a 'surprise' to the target model should mostly come from a non-member document rather than a difficult one. Thus, the reference model should be able to measure the difficulty of a document to the target model with a similar training data distribution. Due to the size and complexity of the pretraining data of LMs, it is difficult to find two pretraining datasets with similar composition, resulting in different data distributions. In order to find models that can reflect the training distribution of the target model, choices include using models from the same family or other models training on the same data. LiRA with Pythia models, GPT-Neo models, and OPT models, with the training data for Pythia-12b-deduped (the Pile) included in their training data, significantly improves over the random baseline. GPT-2 models' training data WebCrawl is filtered from CommonCrawl, which is largely disjointed and different from the Pile, and thus GPT-2 models fails to represents the distribution of the target model's training data with the lowest performance.

**Good reference models should have no overlap in training data with the target model.**    Models that were trained on the Pile all have heavy overlap with the target model in training data, but remain good performance on LiRA. This is because smaller models tend to memorize the exact training data less, making them suitable substitutes for reference models with absolutely no overlap in training

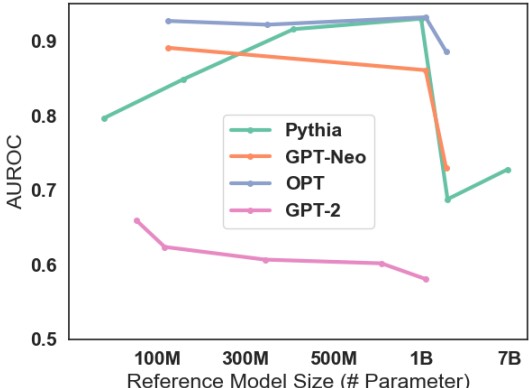

Figure 6: **Effect of the Reference Model.** The Pythia-1b-deduped serves as the best reference model for Pythia-12b-deduped. Using too small or too large reference models hurts the performance of LiRA.

data (Carlini et al., 2021). When we are using models with larger sizes, as shown in Figure 6, LiRA's performance goes down as the reference models memorize more. Also, note that OPT models are more resistant to model size change. This is because OPT models are not exclusively trained on the Pile, making them memorize less.