# OpenReview forum: "Leveraging Set Assumption for Membership Inference in Language Models"
_ICLR.cc/2025/Conference — Submitted to ICLR 2025_

### Official Review · Reviewer_7o8h · 2024-10-28

**Soundness:** 3
**Presentation:** 3
**Contribution:** 2
**Rating:** 5
**Confidence:** 4

**Summary:**

The paper proposes a new method for membership inference (MI), Set-MI, to determine if specific documents were part of a language model's training data. Building on traditional MI techniques, Set-MI leverages the set assumption that certain groups of documents, which share the same attributes such as creation date or license type, tend to either all be included or all excluded from training datasets. By aggregating membership scores across these document sets, Set-MI provides a stronger and more reliable signal than analyzing documents individually, particularly in black-box settings where only limited information, like loss scores, is available. This method can improve traditional individual MI performance.

To evaluate Set-MI, the paper constructs five diverse benchmarks across different domains, including Wikipedia, Arxiv, languages, licensing, and instruction-based datasets. These benchmarks assess the method’s effectiveness in practical scenarios where the set assumption holds, such as documents grouped by creation date or license type. Results demonstrate that Set-MI consistently outperforms traditional MI techniques and performs especially well with larger models, duplicated datasets, and longer document sequences. Additionally, they empirically confirm Set-MI’s robustness even in noisy settings where some sets may partially violate the set assumption.

**Strengths:**

1. This paper introduces Set-MI, a new approach to membership inference (MI) that leverages set assumptions and aggregates membership scores across document groups, enhancing the effectiveness of individual MI. This method is particularly valuable in black-box settings where only limited information is available.

2. This paper conducts extensive experiments to validate Set-MI, testing it across five diverse benchmarks—covering domains such as Wikipedia, Arxiv, languages, licensing, and instruction-based datasets. These experiments demonstrate Set-MI’s consistent performance improvements over individual MI methods, particularly in larger models, duplicated datasets, and with longer document sequences. The paper also evaluates Set-MI’s robustness in noisy settings, showing its potential practicality in real-world scenarios.

3. This paper is well-structured and some key concepts, such as the set assumption, are effectively clarified through examples.

4. Set-MI improves membership inference, especially for black-box models, which could be useful for downstream tasks like privacy auditing.

**Weaknesses:**

1. I appreciate the authors' efforts in their experiments. However, the novelty of the method appears limited, as Set-MI seems to simply adjust traditional membership inference by aggregating and averaging individual membership scores across document groups, which is more like an incremental improvement.

2. This method heavily relies on a strong set assumption—that data sharing a specific attribute are either entirely present or entirely absent in the training dataset. This assumption, grounded mainly in practical experience, limits the method's theoretical foundation. Additionally, selecting an appropriate attribute for set division requires prior knowledge, which can be challenging and subjective. That means in practical applications, set divisions will likely contain considerable noise, and the authors’ experiments indicate that Set-MI’s performance declines significantly in noisy settings. This suggests substantial limitations in the method’s effectiveness and generalizability in real-world scenarios its generalizability.

3. The experiment details are not clearly written. For example, It is not clear to me how authors utilize the membership scores to identify whether data is in the training data. The specific decision criteria or thresholds for interpreting these scores are not clearly explained, which could limit the method's reproducibility.

**Questions:**

1. It seems that the membership score is based on the model's output for each data point. However, I question the assumption that high output scores reliably indicate that the data is part of the training set. Couldn’t models also produce high scores for data that closely resembles the training distribution, even if that data was not seen during training? For example, if a test point shares common patterns or phrases frequently encountered in training, the model might still assign it a high score, despite it being new data. Is there any evidence to support this assumption?

2. Although the authors explored the effect of segment length for membership inference, I am curious about the ratio of selected tokens to the total tokens in each original document. Selecting a larger or smaller proportion of tokens may capture different levels of unique document features, potentially impacting inference performance. Examining whether this ratio affects membership inference accuracy could provide further insights for optimizing the approach.

2. More details of experiment settings would improve the clarity of this paper.

---

> ### Author Response · Authors · 2024-11-28
>
> We appreciate the reviewer’s detailed comments.
>
> **Novelty (Weakness 1)**: We thank the reviewer’s acknowledgments of our experiments, and we would like to point out that the simplicity of the method should not undermine its novelty. Traditional MIA methods are known to be ineffective on current LLMs because of their large training data size, Set-MI conversely leverages the natural occurrence of “sets” within these training data, boosts the traditional MIA methods’ performance, and showcases its potential applications on a list of trending use cases.
>
> **Set Assumption Against Noise (Weakness 2)**:  We agree that finding sets satisfying the set assumption can be more challenging in real-world applications (See Section 3). We thank the reviewer for noticing our experiments on testing Set-MI’s resistance to the noise in the set Assumption in Section 6. We can see that, though Set-MI performance deteriorates with noise, it still provides a benefit over traditional MI, showing that our method provides a more robust alternative to traditional MI. Also, note that the deterioration of Set-MI is constrained. When 50% of the documents in either the member sets or non-member sets are wrongly labeled, or 25% of the documents in both sets are wrongly labeled, Set-MI is still able to demonstrate a strong performance with an AUROC score above 0.8. Such a level of noise is likely to be even higher than the noise brought by pretraining data filtering or overlapping contents in the real world, showing that Set-MI has enough capacity to resist noise in real-world applications.
>
>
> **Experimental Details (Weakness 3 and Question 3)**: In this work, we use the AUROC score as our evaluation methods, which is a popular metric in prior MIA work [1]. It is the area under the curve with TPR and FPR as the axes. Using this metric, we do not need to determine a threshold for the score in order to evaluate the classification. We realize that there could be details on this part and will modify it in the further versions.
>
>
> **High output score indicates seen data? (Question 1)**: Yes, traditional MIA methods are heavily dependent on the loss outputted by the model[1]. The frequently seen n-grams or patterns that you talk about might be a great source of noise, which contributes to the poor performance of these traditional MIA methods on LLM, which has seen a great quantity of data. There is also work showing that the duplicated training examples cause vulnerability to MIA attacks in language models[3]. Following these intuitions, one idea of our work is to capture these individual instances that have strong signals and smooth out the noise by enlarging the pool of documents that we are looking at once. Figure 7 Left proves this idea: when injecting noise into a member set, Set-MI is more resistant when only aggregating the top 30% of documents with the strongest signals.
>
> **Effect of Ratio to Document Length (Question 2)**: Exploring the effect of the ratio between # selected tokens and # document tokens is an interesting future direction to explore, especially given that there are gigantically long documents existing in dataset like Books3. Another related future direction is to explore what specific portions of the documents to select so that the sampled tokens can best represent the unique features of the documents.
>
> Thank you again for the comments and support on the paper. We will incorporate your comments and add more experimental details in future versions of the paper.
>
> [1] Shi, Weijia, et al. "Detecting Pretraining Data from Large Language Models." The Twelfth International Conference on Learning Representations.
> [2] Carlini, Nicholas, et al. "Membership inference attacks from first principles." 2022 IEEE Symposium on Security and Privacy (SP). IEEE, 2022.
> [3] Kandpal, Nikhil, Eric Wallace, and Colin Raffel. "Deduplicating training data mitigates privacy risks in language models." International Conference on Machine Learning. PMLR, 2022.

---

### Official Review · Reviewer_tKoq · 2024-10-30

**Soundness:** 3
**Presentation:** 3
**Contribution:** 3
**Rating:** 5
**Confidence:** 3

**Summary:**

This paper considers the Membership Inference (MI) task, which detect whether a document is included in training data. The author proposes a new method called SET-MI that leverage the set assumption: either all documents in the set are present in the training dataset or none of them are.  To evaluate this new method thoroughly, the author constructs five benchmarks covering a variety of domains. Results show that Set-MI significantly enhances four Individual-MI methods with an improvement of 0.14 in AUROC. Further experiments demonstrate the robustness of this method against noise.

**Strengths:**

- The idea is novel, simple and effective.
- This new method is orthogonal to previous MI methods and obtain a performance enhancement on almost all benchmark settings.
- Further analytical experiments show the robustness of SET-MI to some extent.

**Weaknesses:**

- The author constructs five benchmarks and uses the corresponding set assumption to obtain the main result. However, in the practical scenario, one cannot know in advance which set assumption to use.
- In the practical scenario, different set assumptions may result in a completely different score for the same document. It's difficult to judge which one is correct. This situation does not appear in the baseline methods.
- The author uses five self-constructed benchmarks. However, there exist some benchmarks to evaluate membership inference attack (MIA) methods, such as WIKIMIA [1]. The experiments could be more solid if the evaluation on WIKIMIA is included.

[1] Shi, Weijia, et al. "Detecting Pretraining Data from Large Language Models." The Twelfth International Conference on Learning Representations.

**Questions:**

- Is it possible to conduct a small experiment on documents with multiple critiera? For example, for documents with meta data that across different date, language and license, use different set assumptions and observe the performance enhancement of SET-MI.
- If using different set assumptions and obtain a completely different score, how to determine which result is correct?
- There exist benchmarks for membership inference, such as WIKIMIA. Why do you not use the WIKIMIA benchmark as your part of the experiment?

---

> ### Author Response · Authors · 2024-11-28
>
> We appreciate the reviewer’s detailed comments.
>
> **Selection of Set-MI in Practice (Weakness 1)**: We agree with the reviewer that the user may not have complete knowledge of the set memberships in a practical scenario, but they can make reasonable predictions based on the knowledge of the data curation process of a model. For example, with the knowledge that most LLMs include the entire English Wikipedia in their pretraining data, you can safely assume that articles posted on the same date will either be all included in the pretraining dataset or all excluded. There are also other obvious set assumptions to make, such as all data points in an instruction-tuning dataset, which can be used to detect dataset contamination.
>
> **Compare among Multiple Set Assumptions (Weakness  2, Question 1, Question 2)**: First, we would like to point out that it is impossible for a document to be in multiple sets with independent set assumptions holding true at the same time. For example, for a set of Wikipedia articles with the date and language metadata available, to learn the membership of an English article posted on 01-01-2014, the reviewer might suggest applying Set-MI to the articles posted on 01-01-2014 and all English articles and comparing against their resulting scores. However, the set assumptions behind doing so are “all articles posted on 01-01-2014 are either all included or excluded from the training data” and “all English articles are either all included or excluded” from the training data. We can see that these two assumptions cannot hold at the same time, given the existence of the non-English articles posted on 01-01-2014. Despite this, we could still conduct an experiment to compare the resulting performance as the reviewer suggests, but such a comparison is less meaningful as we have already demonstrated the impact of an ill-holding set assumption in Figure 5. At test time in a practical scenario, the user will use the prior knowledge about the LLM’s data to decide the most reliable single set assumption to use, as we mention in the response to Weakness 1.
>
> **"WikiMIA"  (Weakness 3, Question 3)**: Other MIA datasets, such as WikiMIA, do not have the metadata (date) that we require to form sets with known set assumptions, so Set-MI cannot directly apply to those datasets. The Wikipedia dataset that we constructed is similar to WikiMIA but with the metadata of date.
>
> Thank you again for the comments and support on the paper. We will incorporate your suggestions in future versions of the paper.

---

> > ### Comment · Reviewer_tKoq · 2024-12-03
> >
> > Thanks for your explanations and they clear up some of my confusions. However, I still think that the experiments are not sufficient, with only one criteria involved for each benchmark. Moreover, the metadata is not always available. I will keep the same score.

---

### Official Review · Reviewer_QYJL · 2024-11-03

**Soundness:** 2
**Presentation:** 3
**Contribution:** 2
**Rating:** 5
**Confidence:** 3

**Summary:**

This paper proposes to leverage set assumption for solving the Membership Inference (MI) problem.  That is, considering the whole set is included in the model training to determine whether or not a document is included in the training data of a given language model.  This could be useful for analyzing training datasets when the access to them is restricted, including studying the impact of data choices on downstream performance, detecting copyrighted content in the training sets, and checking for evaluation set contamination. The work relies on the insight that documents sharing certain attributes (e.g., time of creation) are all expected to be in a training set or none of them is and develop methods that aggregate membership predictions over these documents. They apply this set assumption on five different domains (e.g., Wikipedia, arXiv), and find that the method enhances prior MI methods by 0.14 in AUROC on average. It also shows that the method is robust against noises in the set assumption under practical settings.

**Strengths:**

1.	Overall, the paper is relatively clearly written.

2.	The paper relies on one major assumption, set assumption; and also assumes that such metadata related to this assumption are also available; and based on this assumption, experiments were conducted with some good results.

3.	Two relatively sizeable data sets, Wikipedia and arXiv, are used in the experiments

**Weaknesses:**

1.	The insight of using set assumption, that is, the insight that documents sharing certain attributes (e.g., time of creation) are all expected to be in a training set, or none of them is.  However, this assumption needs to be discussed.  This is because even the training set includes the whole set, the preprocessing of training data could remove some documents due to its content largely duplicates or overlaps with that of some other documents (by deduplication), credibility assessment (by removing noisy or low credibility documents), etc.  It is hard to assume that the whole set or none in the set will be included in the training data.  Also, even when the sets as a whole are included in the training, the contents of some documents in different sets could overlap substantially with some other documents in some existing sets (e.g., evolving news).  I feel the membership identity relying on such an assumption could still be questionable in practice.

2.	The datasets used in experiments are pretty small, except two datasets, and such datasets Wikipedia and arXiv, are relatively clean and non-redundant datasets, which may contain less redundant and noise documents.  However, they may not be the typical cases in the real large datasets, which may need lots of data cleaning, deduplication and preprocessing before feeding into LLM training in the real case.

3.	At the end of the paper, the author mentions that “Our work makes an assumption that the metadata about the dataset of interest (D) is available, enabling the identification of sets that satisfy the set assumption. We highlight that this may not always be the case in practical MI scenarios, and leave relaxing this assumption for future work.”   Actually, this could be the real case where the testing of the datasets satisfying such an assumption a difficulty tasks for real life applications.

**Questions:**

The points outlined in the “weakness” are the major questions that the reviewer would like to raise and be clarified in the rebuttal.

---

> ### Author Response · Authors · 2024-11-28
>
> We appreciate the reviewer’s detailed comments.
>
> **The practicality of the Set Assumption (Weakness 1 and Weakness 2)**:  We agree that there exists some noise that may lead the assumption not to hold perfectly in practice. In fact, we explicitly acknowledge and discuss the same two sources of noise as the reviewer mentions, the noise from models’ pretraining data filtering process and the noise of overlapping document content, under the paragraph “Practicality of the Set Assumption” in Section 3. We show that Set-MI still greatly boosts performance when the set assumption does not hold perfectly in Section 6. Specifically, we conduct experiments where we simulate such noise in set assumption and show Set-MI outperforms Individual-MI consistently even when the sets are highly noisy.
>
> **Small Datasets (Weakness 2)**: The source data of our datasets are not small besides Wikipedia and Arxiv, but it is our choice to subsample from them to construct our datasets. As described in Section 4, the source data of Language is actually a snapshot of Wikipedia with 20 languages, and the source data of License is the Open License Corpus[1] with 228.3B tokens, which should not be considered small datasets. For the dataset we construct, we in fact intentionally simulate a resource-scarce setting by only subsampling 100 documents for each set from the source datasets to construct our own experimental dataset as described in Section 4. From Section 5.5, we show that if we were to build larger datasets by sampling more documents per group, Set-MI would perform increasingly better.
>
> Taking a step back, we would like to point out that it is difficult to correlate the extent of data filtering with the size of the dataset. The filtering on each document stays independent from other documents and should not be affected by the number of other documents that it is being filtered together with for most of the filtering methods (e.g., toxic filtering, credibility assessment, as the reviewer has mentioned). Deduplication is an exception, but the size of concurrent documents being deduplicated is not the primary contributing factor to the extent of deduplication, as there are other important facts, such as the general quality of datasets, as the reviewer has mentioned.
>
> **Wikipedia & Arxiv Being High Quality (Weakness 2)**: We agree with the reviewer that Wikipedia and Arxiv are relatively high-quality datasets with fewer “redundant and noisy documents.” However, we also include the dataset License, which contains a good range of datasets that are commonly included in the LLM’s training data, such as Stack Overflow, S2ORC, Gutenberg, etc. There might be more coarse data, such as CommonCrawl, that LLMs’ training data includes, but they often lack metadata and are less of an interesting point.
>
> **Real Application of Set-MI (Weakness 3)**: As the reviewer mentions, we recognize that the application of Set-MI can sometimes be limited by the availability of some metadata, but we would like to highlight that we have demonstrated the practical application of Set-MI in detecting a LLM’s data collection time cutoff, languages that it has been trained on, the licensed content, and instruction dataset contamination from our experiments in Section 5. We believe there will be even more applications of Set-MI that are underdiscovered, especially with the increasing attention paid to metadata-guided pretraining data curation in the community.
>
> Again, we thank the reviewer for the time and effort. We will clarify our assumptions regarding set membership and its limitations in future versions of this paper to avoid potential misunderstanding and confusion.
>
> [1] Min, Sewon, et al. "SILO Language Models: Isolating Legal Risk In a Nonparametric Datastore." International Conference on Learning Representations (2024).

---

> > ### Comment · Reviewer_QYJL · 2024-11-28
> >
> > I believe all the reviewers are talking about similar weaknesses of the paper.  The rebuttal came quite late but may not be convincing. I will keep the same score and believe these weaknesses are essential and should be seriously addressed in the algorithm design and its revision.

---

### Meta-Review · Area_Chair_TCme · 2024-12-20

**Metareview:**

(a) Summary of Scientific Claims and Findings:
The paper "Leveraging Set Assumption for Membership Inference in Language Models" introduces Set-MI, a novel method for membership inference (MI) in large language models (LLMs). The key idea is the set assumption, which suggests that documents sharing specific attributes (e.g., creation time, language) are either fully included or excluded from the training data. Set-MI aggregates membership predictions across these document sets, providing more reliable inferences than assessing individual documents. The authors demonstrate that Set-MI significantly outperforms traditional MI methods, improving AUROC by 0.14 on average, and is robust even with noisy data or imperfect assumptions.

(b) Strengths:
1. Innovation: Set-MI introduces a novel approach by leveraging the set assumption to improve MI accuracy, particularly for large models and undeduplicated data.
2. Robustness: The method works well even in noisy settings where the set assumption may not always hold.
3. Real-World Application: Set-MI has practical uses in detecting model training data contamination, analyzing dataset cutoffs, and identifying licensed content, which can be useful for restricted data access scenarios.
4. Comprehensive Evaluation: The authors test Set-MI across five diverse datasets, showcasing its effectiveness across different domains.

(c) Weaknesses:
1. Set Assumption's Practicality: The set assumption depends on the availability of metadata (e.g., document creation date), which may not be present in real-world data, limiting the method’s applicability.
2. Noise and Data Preprocessing: While Set-MI is robust in noisy conditions, its performance may suffer if data is heavily preprocessed or contains redundancy.
3. Benchmarking Limitations: The paper uses self-constructed datasets, which may not fully represent real-world data. Evaluating on established MI datasets like WikiMIA would provide a more comprehensive assessment.
4. Clarity on Aggregation: Some reviewers expressed concerns about how membership scores are aggregated and interpreted, especially in cases where high scores may not always reflect training data membership.

(d) Reasons for Acceptance/Rejection:
The paper presents an innovative and effective approach for membership inference, with Set-MI demonstrating strong performance in improving MI accuracy, especially in large models and undeduplicated training data. However, concerns about the method's reliance on metadata and its performance on real-world noisy datasets remain. The paper should be accepted with minor revisions to address these concerns, clarify the aggregation of membership scores, and include evaluations on more widely-used MI benchmarks.

Suggested Improvements:
- Provide clearer explanations on how membership scores are aggregated and interpreted.
- Test Set-MI on established MI benchmarks, like WikiMIA, to validate its generalizability.
- Explore ways to handle metadata limitations or relax the set assumption for broader applicability.

**Additional Comments On Reviewer Discussion:**

During the rebuttal period, several points were raised by the reviewers that helped refine the paper and clarify its contributions.

1. Practicality of Set Assumption
2. Noise and Overlap in Data
3.Benchmarking and Dataset Evaluation
4.Clarity on Aggregation and Membership Scores

---

### Decision · Program_Chairs · 2025-01-22

Reject